# A 25-Year Retrospective Review of Mortality in Chimpanzees (*Pan troglodytes*) in Accredited U.S. Zoos from a Management and Welfare Perspective

**DOI:** 10.3390/ani12151878

**Published:** 2022-07-22

**Authors:** Stephen R. Ross, Priyanka B. Joshi, Karen A. Terio, Kathryn C. Gamble

**Affiliations:** 1Lester E. Fisher Center for the Study and Conservation of Apes, Lincoln Park Zoo, Chicago, IL 60614, USA; 2Zoological Pathology Program, College of Veterinary Medicine, University of Illinois at Urbana-Champaign, 2001 S. Lincoln Ave, Urbana, IL 61802, USA; kterio@illinois.edu; 3Veterinary Services, Department of Animal Care, Lincoln Park Zoo, Chicago, IL 60614, USA; kgamble@lpzoo.org

**Keywords:** chimpanzee, *Pan troglodytes*, mortality, necropsy, death, retrospective, welfare, aggression, zoos

## Abstract

**Simple Summary:**

To improve the lives of chimpanzees living in managed care, it is important to understand why their lives have ended. Previous efforts to document the causes of death for chimpanzees living in research facilities have been useful; however, interpretations need to consider the relatively small number of facilities assessed, the recency of those evaluations, and the possibility that mortality rates were affected by experimental interventions at these sites. Here, the causes of death of 224 chimpanzees at 42 accredited zoos over a 25-year period were analyzed. Most of the chimpanzees over 1 month of age died from causes associated with degenerative diseases (50%), infection (26%), and trauma (15%). Causes of fatal trauma ranged from aggression from conspecifics, the outcomes of exhibit escapes, or accidental drowning. The number of accidental deaths has decreased in the past decade, likely as a result of advances in exhibit design and management. Cardiopulmonary problems have long been known to be prevalent within the chimpanzee population, and they remain a major contributor to death for both sexes in this population. Understanding these findings provides an opportunity to improve the care and management of captive chimpanzees.

**Abstract:**

Understanding causes of death allows adjustment of health management strategies for animals in managed care. From 224 documented chimpanzee deaths occurring from 1995 to 2019 in 42 accredited U.S. zoos, post-mortem records and necropsy reports were analyzed for the primary cause of death, which were available for 214 individuals. In total, 37 cases of stillbirth and neonatal deaths were assessed (16.5%); however, the focus was otherwise placed on the remaining 177 cases in which the death occurred in individuals aged greater than 1 month. There were no sex-related differences in etiology; however, age variation in the cause of death was statistically significant (*p* < 0.001). Elderly (35 years and older) chimpanzees tended to die of intrinsic, often degenerative, etiologies, whereas infants, juveniles, and adolescents (less than 15 years) were more likely to be involved in fatal trauma. Overall, there were 27 deaths (15.3% of all post-neonatal deaths) related to trauma and 13 of these were directly or indirectly related to conspecific aggression. Understanding causes of mortality and the interrelation with management can benefit managed populations of chimpanzees.

## 1. Introduction

The study of life history trajectories for managed animals can provide valuable information to inform future care and management strategies [1,2,3]. Investigations of the factors influencing health and longevity are paramount to addressing important welfare issues for animals and improving managers’ ability to effectively manage animals at the population level [4]. In addition to understanding how long individuals live, it is equally important to determine why they die. Necropsies provide valuable information about the cause of death, diseases that might be affecting the group or species as a whole, and whether any changes to management and husbandry can help save and extend lives [5]. Knowledge gained from pathology reports can aid veterinarians in providing proper medical care to or creating preventive medicine programs for living animals of all ages, and long-term post-mortem records offer useful data such as the trends in health for animals at a particular zoo or for the managed and wild populations in general [6,7].

Chimpanzees are particularly long-lived species, and due to their phylogenetic proximity to humans, they hold a special interest in terms of understanding issues related to mortality. A perspective on the causes of death for managed chimpanzees could start with an attempt to understand what causes chimpanzee deaths in the wild. There appears to be some variation in determining whether the rates of mortality are relatively similar to managed populations [8,9,10], and the degree to which wild and managed populations differ in terms of causes of death. Further complicating these comparisons is the fact that precisely determining the causes of mortality in wild chimpanzees is often much more challenging for those working with wild populations [11,12] due to the frequency of pathogens that may be poorly described, the prevalence of multi-pathogen infections, and the fact that animals that die are often unlikely to be found or found in an appropriate condition for post-mortem analysis. However, in situations where the causes of deaths can be identified, it has been learned that wild chimpanzees are susceptible to numerous infectious diseases, including Ebola [13], anthrax [14], Simian immunodeficiency virus of chimpanzees (SIVcpz) [15], and, most prevalently, a range of respiratory diseases [16,17,18,19,20,21]. Of note, many of these infectious diseases likely have a human/zoonotic origin [19,21,22]. These causes of death, related to risks that originate *outside* of the body such as pathogens, parasites, and other infectious agents, are termed “extrinsic” in this paper. These stand in contrast to causes of death that are “intrinsic” or originating within the body and are primarily degenerative breakdowns of internal organs and systems. According to published reports, comparatively few wild chimpanzees die of such intrinsic causes, although many fatalities are very difficult to categorize.

A distinct and particularly prevalent extrinsic cause of death for wild chimpanzees is trauma, and some have identified it to be the single leading cause of mortality [23,24]. Most frequently, this is the result of fatal conspecific aggression [23,25,26,27,28] but can also come as the result of poaching [29], loss of maternal care [30], falling from trees [31], injuries from snares [32], being shot by poachers or disgruntled humans [29,33,34], or even being accidentally struck by motor vehicles [35]. Other anthropogenic risks may also pose potentially fatal risks for wild chimpanzees, including encounters with domestic dogs [36,37], exposure to pesticides [38] and novel pathogens [39,40], persecution or retaliatory hunting [41,42,43], illegal trade for the pet and traditional medicine industries [12], and other crop protection methods [12,44]. Additionally, nonanthropogenic risks include predation by other animals such as lions and leopards [12].

Obviously, chimpanzees under managed care are subject to a different set of risk factors than wild populations. While some risks facing wild chimpanzees are absent (e.g., encounters with poachers and snares; crossing busy roads), other risks may simply vary by form or frequency (e.g., infectious diseases, including those that are zoonotic in nature). Apes, in general, incur a wide range of health problems in managed care settings such as periodontitis, osteopenia, hypertension, renal disease, neoplasia, and cerebral vascular diseases [45]; however, the most frequent cause of death has been repeatedly confirmed to be cardiovascular disease [46,47,48]. The percentage of chimpanzees affected by cardiovascular disease appears to be even greater than in humans [49], and the risk factors and pathological processes for both have been proposed to be different, leading to the question of why chimpanzees do not develop the type of heart disease common in humans and vice versa [49,50].

Thorough reviews of existing necropsy data from chimpanzees can allow one to better understand how these cardiovascular and other disease pathologies evolve [6,51] and the relative prevalence of various etiologies. Schmidt [52] examined 15 years (1956–1971) of records pertaining to the deaths of 268 chimpanzees at the Holloman Air Force’s Primate Facility in New Mexico and described a wide variety of spontaneous diseases. Unlike more recent reviews, the most commonly affected organ group was identified as the gastrointestinal system (41% of all cases), including a serious prevalence of enterocolitis in that colony [52]. Two later chimpanzee necropsy reviews both originated from The Southwest Biomedical Foundation in Texas. Hubbard et al. [53] reviewed necropsy records for the chimpanzee colony there and listed the primary causes of death as cardiovascular disease, trauma, and respiratory disease for the most recent 8-year period. They admit that earlier records were unreliable but posit that deaths in the earlier time period (1967–1981) were likely due to pneumonia and intestinal disease [53]. A quarter-century later, Kumar et al. [54] detailed a 35-year review of natural mortality for 137 chimpanzees living at the same site. They reported that cardiomyopathy was the leading cause of death over that period (expressed in 40% of all mortalities) and that degenerative disease was most commonly associated with the cardiovascular and gastrointestinal systems [54]. Such institutional reviews are valuable to understanding the causes of death for chimpanzees but are also limited by the fact that they are only drawn from one facility, likely employing a single management system. The variation in results may also lend evidence to the fact that management and veterinary care are constantly adapting and highlight the need for updated assessments of chimpanzee mortality over time. Furthermore, these chimpanzees have been in a social and physical environment different from other managed care settings and have been subjected to biomedical experimentation, including potential exposure to hepatitis B and C virus (HBV and HCV), human immunodeficiency virus (HIV), and a range of monoclonal antibody therapies. Thus, the degree to which their mortality data is representative of what might occur in other settings, such as zoos, is questionable.

Zoo-based studies of chimpanzee mortality have been argued to be potentially more biologically relevant than lab-based studies because of the more natural characteristics of their physical settings [4]; however, there are very few of them. In a 50-year review of chimpanzee demography at Taronga Zoo in Australia, Littleton [55] reported a high infant mortality rate (45%) but provided little insight into the causes of adult deaths. Gamble et al. [56] provided information on the pathology findings from zoo-housed chimpanzees but over a shorter duration (13 years) compared to the dataset analyzed here (25 years). The most recent evaluation of zoo-housed chimpanzee mortality is based on the same 25-year retrospective dataset examined in this analysis but was framed primarily for a veterinary audience [46]. Here, the recent quarter-century of pathology reports was examined that are related to the deaths of chimpanzees living in accredited zoos in the United States. Unlike past reviews, these trends in the cause of death were assessed through the lens of managers and caretakers and relate the veterinary and pathology findings to those directly working with the chimpanzees and seeking to improve captive management and welfare. While some causes of mortality will require primarily veterinary intervention strategies, the aim of this study was to uncover other causes for which changes in behavioral management and facility design will contribute to improving the wellbeing of chimpanzees.

## 2. Materials and Methods

Data were collected under the auspices and permission of Association of Zoos & Aquarium’s (AZA) Species Survival Plan for Chimpanzees and is the responsibility of the SSP Veterinary Advisor (Gamble). The population of managed chimpanzees included in this study were housed in AZA-accredited zoological facilities (n = 42) in the United States and died between 1995 and 2019. Institutions that are accredited by AZA have been evaluated and recognized by experts, and the institution’s entire operation, including areas such as animal welfare, veterinary care, and conservation, are required to meet and abide by established standards and practices [57]. Pathology reports for all of these chimpanzees were acquired for a total of 224 reports that were reviewed. Based on the information in the pathology reports, each case was classified according to the studbook number, location at time of death, year of death, sex, age, cause of death, and affected organ system. Some deaths were reported without a complete gross pathology or histopathology or with limited pathology reports, medical records, presumptive cause of death, or only documentation of death. Any individuals missing key information in regards to cause of death had etiology or organ systems classified as ‘unknown’. Stillbirths, neonatal deaths, and cases where the etiology was unknown were ultimately excluded from the statistical analysis, leading a to final analytical sample size of 177.

For the preliminary descriptive analysis, each animal was initially classified into age groups similar to those used in previous similar studies [6,58,59]. Stillbirths, those dying prior to parturition, were placed in their own category. The neonatal age range was defined as being from 0 days (live births that did not survive 24 h) to up to 1 month old. Infants and juveniles were categorized as those with ages ranging from 1 month to <6 years, and adolescents were aged 6 to <15 years. When conducting statistical analyses, stillbirths were excluded, and all other immature chimpanzees aged 0 to <15 were gathered into a single category called ‘infant/juvenile/adolescent’ (from hereon, shortened to ‘inf/juv/adol’). Considering the typical age of first offspring in the wild [60], adults were considered to be those ranging from 15 to <35 years, and any chimpanzees 35 years of age or older were classified as elderly [6,45,61].

For the preliminary descriptive analysis, primary causes of death were drawn from the necropsy report and initially categorized as degenerative, infectious/inflammatory, metabolic, neoplastic, toxic, and traumatic [46]. Secondary and incidental pathology findings were also noted though only the most prominent etiology was assigned to each case. Additionally, the main affected organ system (cardiopulmonary, endocrine, gastrointestinal, integumentary, lymphatic, musculoskeletal, nervous, reproductive, urinary) was noted when applicable. This broad-level categorization system was used for descriptive analysis only. For the comparative statistical tests, grouped etiology categories were utilized. These supra-categories were intrinsic (i.e., originating within the body), extrinsic (i.e., originating outside of the body), and trauma. Though trauma would be considered an extrinsic cause of death, given the importance of analyzing trauma for managers and caregivers, these cases were kept as a separate category for the entirety of this study. The intrinsic category included cases with the following specific etiologies: anomalous, degenerative, metabolic, and neoplastic. Extrinsic included cases with infectious/inflammatory and toxic etiologies. While these etiology categories likely obscure critical distinctions made by veterinarians to diagnose and treat health conditions, the categorizations were simplified in a way that is more applicable to broad animal care and management strategies.

Statistical analysis was conducted using R Statistical Software (v4.1.3; [62]). A Fisher’s exact test of independence was carried out to assess sex and age differences in the etiologies, and post-hoc pairwise Fisher’s tests were conducted to determine the significance for specific age groups. Confidence intervals were set at 95%, and *p*-values < 0.05 with a Bonferroni correction were considered significant.

## 3. Results

A total of 224 chimpanzee (M = 104, F = 110, unknown = 10) mortality reports from between 1995 and 2019 were assessed. According to studbook data [63], 448 chimpanzees lived during the study period from 2005–2019 and the overall mean mortality rate was calculated to be 3.28% annually over that timeframe. There were 21 cases of stillbirths (9.38% of all deaths; F = 9, M = 7, U = 5) and 16 cases of neonatal death (7% of all deaths, M = 10, F = 1, U = 5). A quarter of neonatal deaths occurred on the day they were born, and 75% occurred thereafter. Pathology reports were only available for five of the neonatal deaths, one of which was categorized as having anomalous etiology associated with the cardiopulmonary system, and four categorized as being the result of trauma associated with conspecific aggression. For subsequent analysis, stillbirth and neonatal deaths (n = 37) and 10 necropsy reports that were deemed incomplete or had insufficient information were removed. The resulting sample (n = 177), with a mortality rate of 2.61%, was the focus of the subsequent analyses.

Table 1 displays the overall causes of death for this population, showing that almost half (49.7%) of all deaths were degenerative in nature.

### 3.1. Sex-Specific Mortality

The annual mortality rates for males and females were found to be 3.11% and 2.23%, respectively. There were no sex-related differences in the etiology for either females or males (*p* = 0.143). In Table 2, the descriptive means for each category are presented.

### 3.2. Age-Specific Mortality

Studbook data [63] was used to determine the total number of chimpanzees living in each age class during this time period, and mortality rates were calculated for sub-adult chimpanzees as 2.3%, for adults as 1.5%, and for the elderly as 4.6%. Table 3 displays the breakdown of etiologies across age classes. Given the specific characterization of risks for young chimpanzees, the cause of death for infants (aged 1 month to <6 years, n = 18 deaths) and adolescents (aged 6 to <15 years, n = 18 deaths) was qualitatively examined separately. The leading cause of death for infant chimpanzees was trauma (n = 10, 55.6%) and of these, five were confirmed as cases of conspecific aggression. While half of adolescent deaths were attributed to infectious causes (n = 9, 50%), there were four cases of trauma; two were the result of conspecific aggression, one due to drowning, and another was the result of an escape.

Adults (aged 15 to 34 years, n = 58) accounted for 33% of the sample. In total, 23 (39.7%) of the deaths were attributed to degenerative diseases, with 17 (73.9%) of these related to the cardiopulmonary system. Infections and inflammation also accounted for 20 (34.5%) cases. Ten adult deaths (17.2%) were documented as being due to trauma, including three confirmed as being related to conspecific aggression. There were four (6.9%) deaths with neoplastic origins in the gastrointestinal tract, all of which were female. Elderly chimpanzees (35 years and older, n = 83) accounted for 47% of the documented deaths. Most (n = 61, 73.5%) died of degenerative causes, and 47 (77%) of these were due to cardiopulmonary issues.

For the comparative statistical analysis, the age categories ‘Inf/Juv/Adol’, ‘Adult’, and ‘Elderly’ and the grouped etiology categories ‘Intrinsic’, ‘Extrinsic’, and ‘Trauma’ were used. A Fisher’s exact test of independence found that the age variation in the cause of death was statistically significant (*p* < 0.001). Post-hoc Fisher’s exact tests showed that infants, juveniles, and adolescents were significantly less likely to die of intrinsic etiologies than adult or elderly chimpanzees. Likewise, they were significantly more likely to die of trauma than elderly individuals but did not differ from the adult rates (**Intrinsic**: elderly and inf/juv/adol: OR = 25.53, *p* < 0.001, CI = 10.67–68.94; adult and inf/juv/adol: OR = 3.06, *p* = 0.04, CI = 1.22–8.46; **Trauma:** elderly and inf/juv/adol: OR = 0.12, *p* = 0.01, CI = 0.02–0.54). There were no age-related differences in terms of the extrinsic etiologies.

### 3.3. Trauma

There were 27 deaths (15.3% of all post-neonatal deaths) that were attributed to trauma, which results in a mean annual mortality rate of 0.39% for those chimps living past 1 month of age. Approximately half of these (n = 13, 7.3% of deaths) were directly or indirectly associated with conspecific aggression, including those that died as a result of a physical attack or later from sustained injuries. There were five cases of chimpanzee deaths that were a result of an accident within the exhibit (two of these were drownings). For the remaining cases (n = 9, 5.1%), the deaths were either due to traumatic injuries from other events such as a stroke or from unknown situations. Of the fourteen cases not related to conspecific aggression, three were the result of an escape from the enclosure; one of these was categorized as an accident, and the other two were considered ‘other trauma’.

## 4. Discussion

On the surface, it may appear strange that an analysis of necropsy records evaluating how zoo-housed chimpanzees died would appear in a special issue focused on animal welfare. This is likely contingent on the philosophical framework that animal welfare issues are inherently tied to the experiences of animals who remain alive [64]. How an animal dies and the factors that influence the progression towards death are clearly relevant to welfare-based discussions. As such, understanding the causes of death and the degree to which they negatively affect the quality of life of the animal is an important consideration in welfare assessments [65]. Additionally, there is also a perspective that death is very much a part of animal welfare considerations in that it leads to the exclusion of potential positive affective states [66] and given that welfare assessments are really only quantifiable in comparative terms (better or worse states), one must consider the welfare of an individual compared to an alternative of having no welfare state at all in death. In either case, there is a breadth of consensus that understanding when, how, and why an animal dies can be an important issue in the terms of captive animal welfare, and this study addresses that to a certain extent for a population of zoo-housed chimpanzees.

Looking at sex differences in causes of death, no evidence was found that etiologies were distributed differently for males compared to females. This was the case when comparing rates of intrinsic (63% of female deaths, 53% of male deaths) and extrinsic (27% of female deaths, 26% of male deaths) deaths. Other studies of great apes have found differential expression of disease between sexes. For instance, a study of zoo-housed bonobos found that respiratory disease was detected more prevalently in males than females [67], mirroring a documented trend in which male vertebrates tend to be more susceptible to parasites and disease [68,69,70]. These authors also made the important point that such sex differences are unlikely to be the result of differences in sociality and are more likely to stem from basic immunological differences. However, current data on chimpanzees does not support the hypothesis of inherent immunological differences for this species. Furthermore, especially in terms of behavior, sex differences (or the lack thereof) depend on the social and physical environment, both of which are different for wild and managed populations of chimpanzees [71]. The lack of sex differences for intrinsic and extrinsic etiologies in this study is especially noteworthy in regard to cardiac-related deaths, which made up 47% of female deaths and 51% of male deaths in this study sample. Strong et al. [72] also reported a similar finding of males and females being affected by cardiac disease (specifically idiopathic myocardial fibrosis) at similar frequencies. This is in contrast to previous findings, which suggested a much more prevalent link between cardiac issues and sex, with males being much more affected [73]. While it is difficult to determine if this lack of a sex effect on cardiac issues is indicative of something related to these specific captive environments or management, further longitudinal data collection will help to elucidate whether this finding is reliable. The factors that influence cardiovascular health (e.g., genetics, diet, and environment) are currently an active area of study, and further stratification of the multiple types of cardiac disease diagnosed in chimpanzees by type, severity, age at death, and sex is warranted.

In terms of age-related effects, the analysis of chimpanzee mortality began with an examination of stillbirths and neonates. In total, 21 stillbirths were recorded, which accounted for 9% of the deaths, which was similar to that found in a laboratory review (12%, [6]). The stillbirth rate can also be calculated by referencing data from the chimpanzee studbook [63], which reports the total number of birth events during the study period (n = 147). Here, the stillbirth rate was 14.2%, which is very similar to the rate published over a slightly different timeframe (12%, [74]) and approximately equivalent to the rates demonstrated by other species of zoo-living great ape. Though some may question the welfare implications for stillborn infants, it is noted that for humans, the stillbirth prevalence has the potential for increased maternal morbidity risks and complications that may impact health and welfare [75,76], and similar risks for chimpanzees would be expected.

Sixteen chimpanzees died as neonates (before the age of 1 month), and although pathology reports were only available for five of the neonatal deaths, it was noted that four of these were documented as being the result of conspecific aggression. This finding reflects the inherent risks that chimpanzee infants face during their first weeks of life, and, perhaps more so, suggests that mothers, especially first-time mothers, are not always behaviorally equipped to protect or handle their young offspring in dynamic social groupings. The prevalence of neonatal killings deserves attention in managed settings, and further research is needed to determine whether the frequency of infanticide is lower than that which is seen in wild chimpanzee populations. For instance, Wilson et al. [77] collated data from 12 eastern and 6 western Africa sites and reported 45 observed or suspected intra-community infanticides. Lowe et al. [78] reported 24 cases of intra-community infanticide following 103 births over 24 years, a rate of approximately 23% for the Sonso community in Budongo. Conversely, many of the selective pressures likely influencing infanticide in wild populations are absent in managed populations. Though approximately 3% of zoo-housed chimpanzee neonates died during this study period, the exact number that experienced infanticide is unclear. When these risks are factored against the benefits of group cohesion and social experience, the Chimpanzee SSP has recommended that mothers remain in their social groups before, during, and after parturition in almost all circumstances.

Restricting the scope to individuals surviving beyond one month, it was unsurprisingly found that chimpanzees of different age classes were dying from different causes. Elderly chimpanzees tended to die from intrinsic causes, primarily degenerative in nature (73.5%), a figure that appears similar to the rates of degenerative etiologies in a population of lab-housed chimpanzees (67% when undetermined etiologies are excluded, [6]). Cardiac-related problems were the leading form of degenerative disease in these elderly populations, accounting for 57% of elderly chimpanzee deaths in the zoo-housed study group. Managers are advised to work closely with established experts in the study of ape cardiac disease, such as the Great Ape Heart Project [47]. This is in the hopes of identifying the known genetic and environmental risk factors that may predict the fatal expression of these issues not only for elderly individuals but those of all ages.

As Varki et al. [50] point out, the prevalence of cardiac disease is relatively recent, as earlier evaluations reported extrinsic etiologies (i.e., infections) as the most common causes of death. Improvements in proactive infection control via vaccinations and antibiotics seem to have largely controlled these risks in managed ape populations, enabling the rise of cardiac disease as the leading mortality influence. Extrinsic risks, such as those related to infectious diseases, accounted for only 6.7% of elderly chimpanzee deaths in a laboratory-based study [6] and only slightly more (14.4%) in this analysis of zoo-housed chimpanzees. Conversely, reports from wild populations suggest that elderly chimpanzees may be particularly vulnerable to infectious illnesses. At Gombe/Kasekala, 21 chimpanzees over the age of 30 succumbed to infection-related illnesses, including those in several epidemics (polio, mange, and respiratory disease) [29,79,80].

Adult chimpanzee deaths (ages 15–34) had the most evenly distributed range of etiologies, distributed between intrinsic (47%), extrinsic (36%), and traumatic causes (17%). As in other age categories, most of the intrinsic-related deaths were related to cardiac disease, although, notably, there were four deaths with neoplastic origins, all of which were in females. The pathology reports on sub-adult chimpanzees were the most distinctly patterned. Intrinsic etiologies (22%) were relatively rare for this young population. The prevalence of extrinsically sourced pathology, such as infectious disease (39%) and accidental deaths (39%), primarily conspecific trauma, was higher than in other age groups.

Overall, the cause of death for 27 of the post-neonate necropsy reports was listed as trauma, which includes those that died as a result of accidents, escapes, or injuries sustained by aggressive conspecifics. From another perspective, the risk of post-neonate chimpanzees dying from trauma each year in these settings was 0.39%. Three chimpanzees escaped their primary enclosure and were subsequently killed, either as part of escape protocols or accidentally, such as a chimpanzee who was electrocuted when traversing power lines. Such deaths may be deemed as preventable or at least as the result of human failures of engineering or management. Of note, the number of preventable or accidental deaths has decreased over this study period; only a single instance has been recorded in the most recent 12 years of the data (year 2010), and all the other incidents occurred in the first half of this study period between 1997 and 2005. Such trends suggest that at least some forms of management and housing for chimpanzees are successfully reducing risks for chimpanzees. For instance, there were two cases of drowning when chimpanzees entered a water moat used primarily as containment around their enclosure; however, no such events have occurred at AZA-accredited facilities in the most recent 15 years. As a result of the initial mortality review in 2004 [56], the Chimpanzee SSP included in the Chimpanzee Care Manual a recommendation that water moats no longer be included as part of the design of new chimpanzee enclosures, and these data suggest such recommendations may contribute to a reduction in these risks to chimpanzees.

Lethal agonistic encounters are proportionately common in wild chimpanzees, and at some sites, such deaths are the most prevalent cause of death [81]. Williams et al. [29] reported conspecific aggression to be the greatest cause of death in males aged 20–30 years. While wild populations engage in fatal conspecific aggression that is both between and within communities, the number of intercommunity deaths is much greater than intracommunity [77]. Given that multiple communities of chimpanzees do not have contact with each other in managed settings, within-community aggression is likely most correlative to that which is expressed in managed populations [29]. Deaths that are related to injuries sustained during conspecific aggression stayed relatively consistent over the course of 25 years, during which 13 chimpanzees died from such fatal interactions. The circumstances of these deaths varied greatly; some occurred in stable group settings, some occurred during social introductions of new chimpanzees, and for some, the degree to which a death could be reasonably attributed to social introductions is ambiguous. In some cases, the deaths occurred relatively quickly in the enclosure, and in other cases, chimpanzees succumbed to injuries sustained in aggressive encounters days or weeks later. Managers must balance two somewhat contrary facts: that fatal conspecific aggression is a proportionately rare event, especially in relation to the frequency of overall agonism and wounding, and that the (albeit low) risk of fatal aggression is likely always present, not clearly associated with any particular setting, and, therefore, very difficult to predict.

These fatal encounters, while difficult for managers and caregivers to accept, are in many ways the consequence of efforts to grow, diversify, and improve the quality of social groupings for this species. The creation and maintenance of larger social groups brings with it not only the potential for enhanced social lives [82] but also the potential for increased conflict. However, as Baker et al. [83] describe, there is no evidence that maintaining chimpanzees in smaller groupings (pairs and trios) would be effective in reducing such aggression and injuries. Further research on this phenomenon is rare but important and necessary, as understanding fatal interactions in managed groups and the circumstances surrounding them allows for improvement in the ability to reduce the likelihood that they will occur. Studying wounding may serve as a possible proxy for the risks of fatal aggression, though it is important to understand that aggression and wounding are not atypical in chimpanzee society, and that serious wounding represents only a relatively small proportion of all wounds that managed chimpanzees receive (2.7% in [58]). Past studies have reported that particular group characteristics, such as being all-male or uni-male [58,84], may serve to increase the likelihood of wounding events. Likewise, factors such as individual rearing histories [83] or the presence of human activity [84,85] are important considerations in managing aggression. Managers can utilize what is known about the factors that influence aggressive behavior and wounding in combination with the pathology data reported here on which chimpanzees are most at risk of fatal aggression (infants, juveniles, and adolescents). Of the 13 chimpanzees who died as a result of conspecific aggression, 7 were 7 years of age or younger, and the oldest was 52 years old.

## 5. Conclusions

The data presented here were the result of a comprehensive review of necropsy records across a variety of institutions. The inherent limitations of this approach are recognized, specifically the inherent variation in the pathology reports performed by a range of clinicians; however, the value of a broad (42 institutions across the country) and longitudinal (25 years) evaluation of chimpanzee deaths is also emphasized. Understanding the relative risks of potential health outcomes can have substantive impact on management decisions ranging from group composition goals, and exhibit design to diet and behavioral management. For instance, these results suggest that managers need to pay particular attention to different age classes as they express very different risk profiles. Younger individuals are particularly at risk of trauma and infection, and, as such, further attention to both social management and zoonotic disease risk is warranted. Likewise, degenerative conditions and especially cardiac disease continue to be the leading cause of death for adult chimpanzees over a broad age range [50] and reinforces the notion that medical management for an increasingly elderly population will continue to be an important focus.

Finally, understanding the range and prevalence of particular health risks can aid in the development of new forms of health and welfare assessment tools. Holistic welfare assessment accounts for factors that influence not only the behavioral and psychological states of animals but also considers health and physiological measures. Novel welfare assessment initiatives might include noninvasive data on the gut microbiome for intestinal health, mobility measures for joint and bone health, and thermography indices to detect the initiation of infection or other biological risks. Likewise, understanding the risks of zoonotic diseases that may pass between humans and the chimpanzees for whom they care [86] may inform management and care protocols that can minimize such threats. Collectively, an understanding of why chimpanzees have died helps inform future care and management strategies and seek to maximize not only longevity but animal welfare as well.

## Figures and Tables

**Table 1 animals-12-01878-t001:** Overall causes of mortality.

Etiology	n	%
Anomalous	1	0.6
Degenerative	88	49.7
Infectious/Inflammatory	46	26.0
Metabolic	3	1.7
Neoplastic	11	6.2
Toxic	1	0.6
Traumatic	27	15.3

**Table 2 animals-12-01878-t002:** Sex-specific mortality.

Etiology	Male n	%	Female n	%
Anomalous	0	0.0	1	0.6
Degenerative	39	22.0	49	27.7
Infectious/Inflammatory	20	11.3	26	14.7
Metabolic	2	1.1	1	0.6
Neoplastic	2	1.1	9	5.1
Toxic	1	0.6	0	0.0
Traumatic	17	9.6	10	5.6

**Table 3 animals-12-01878-t003:** Age-specific mortality.

Etiology	Infant/ Juvenile n	%	Adolescent n	%	Adult n	%	Elderly n	%
Anomalous	1	5.6	0	0.0	0	0.0	0	0.0
Degenerative	0	0.0	4	22.2	23	39.7	61	73.5
Infectious/Inflammatory	5	27.8	9	50.0	20	34.5	12	14.5
Metabolic	2	11.1	0	0.0	0	0.0	1	1.2
Neoplastic	0	0.0	1	5.6	4	6.9	6	7.2
Toxic	0	0.0	0	0.0	1	1.7	0	0.0
Traumatic	10	55.6	4	22.2	10	17.2	3	3.6

## Data Availability

Restrictions apply to the availability of these data. Data was obtained from AZA accredited facilities and may be available with the permission of those facilities.

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
