# Peer review of "A 25-Year Retrospective Review of Mortality in Chimpanzees (Pan troglodytes) in Accredited U.S. Zoos from a Management and Welfare Perspective"

_animals, 2022, doi:10.3390/ani12151878_

Round 1

Reviewer 1 Report

Animals - 1779845  A 25-year Retrospective Review of Mortality in Chimpanzees 2 (Pan troglodytes) in Accredited U.S. Zoos from a Management 3 and Welfare Perspective

General:

The submitted manuscript presents a retrospective study of mortality for 224 chimpanzees across 42 accredited zoos with the goal to inform management and improve wellbeing of these animals.  The manuscript is generally well-constructed, organized, clear, and to the point. It was generally easy to read and follow.  The manuscript presents new material, is of moderate impact, and will likely be of interest to the readership.  I agree with the author that a better understanding of mortality can help inform care and management.  I only have one suggestion for clarity, and a couple comments that can be take at the author’s discretion. 

Results:  3.3 Trauma:  Line 240: The last sentence is confusing due to its inclusion in the trauma section; it really sounds like you are talking about more trauma cases (stroke and unknown are not trauma) and I had to stop and figure out that you had switched to a different topic.  I would move the sentence “For the remaining cases (n=8, 4.5%), the deaths were either due to other events such as a stroke or the circumstances surrounding the deaths were unknown.” Into a new subsection 3.4 (call it “Miscellaneous” or some such) even though it is only one sentence.

Optional:

Discussion:  Line 294:  Consider that in addition to not being as protective of her offspring, first time mothers may also provide poor/rough handling and kill/injure the offspring themselves.  Pretty certain that this is an issue based on other species.

Discussion:  Line 305:  Maintaining social grouping may also benefit by exposure to experienced mothers and the potential that the new mother may improve maternal skills in that environment.  This one is pure speculation on my part.

Reviewer 2 Report

This is a very well written paper. It represents a significant body of work, and the findings are of interest not only to the readers of this journal but the wider scientific community. What's more, the findings of this work are clearly relevant to chimpanzee welfare and management, and can be used to inform decision making on an individual zoo and wider population level. The authors do a very good job of explaining the validity of their conclusions valid and their relevance to the industry, whilst maintaining a quality, scientific feel to the paper. There are a few minor points I have raised - see individual comments on the manuscript, that I feel would help improve the paper. Specifically, the separation of "traumatic" and other "extrinsic" causes of death requires some clarity, as does the grouping of deaths according to age and the use of the term "subadult". I have also made some additional suggestions for further discussion points. 

There is no statement of ethical approval. Whilst I have no concerns about the ethics of this study, the inclusion of a statement relating to ethical approval is good practice and is required prior to publication.

Otherwise, I have no hesitation in recommending this manuscript for publication in this journal. I believe it to be of a good standard and of valuable significance.

Reviewer 3 Report

This paper provides an important analysis of chimpanzee mortality in accredited zoos, as they highlight there is relatively little information on this topic and a better understanding of the factors linked to mortality has important implications for captive chimpanzee welfare. They pull together an impressive data set across US zoos that allows

them to disentangle a range of different factors.

I have only minor comments below:

Paragraph lines 73-80. Given its importance in the decline of wild chimpanzee populations, it may be worth explicitly mentioning deaths as the result of collection of young individuals for (typically illegal) sale to the pet, ‘entertainment', or biomedical industries (both conspecifics and the infants themselves). Some populations also experience important (non-human) predation threats - for example, leopard predation in Tai.

Lines 114-118. Is may be worth highlighting that in addition both the social and physical environment are likely very differently structured in a biomedical care setting.

Line 154 - I'd suggest labelling the 0-14yrs chimpanzees as "immature' rather than 'subadult;, to avoid confusion with the wider literature (sub adulthood is typically a defined dew-lopmental period between independence from the mother and full adult development at around 8-16 years old).

Line 174 - could you also provide the version of R being run in Studio (and reference this), as I think that's the appropriate convention.

Results - you mention in the simple summary that the number of accidental deaths has declined in the past decade, but these data are not presented in the results. Perhaps they could be plotted or summarised in a table? (I see that they are noted in line 344-347 of the discussion but could still usefully be outlined in the results).

Discussion

Line 266 relevant to this point, is that there is at best very limited opportunities for captive chimpanzees to express natural sex differences in sociality, given typical enclosures almost all individuals are aware (at least acoustically) of the location (and often activity) of all other individuals. Unlike wild chimpanzees - and to an extent bonobos - that show strong sex differences in sociality, for example in fission-fusion structure.

Line 277 I wonder if there is scope for even speculative comment here on what might impact cardiovascular disease in captive chimpanzees e.g. genetic predisposition, diet, environment.. etc. or what more specific

features the authors may recommend exploring/monitoring in the future to gain a better understanding.

- While these reports are accurate they are particularly biased by one - highly infanticidal - community within the Budongo Forest, in Uganda. Across West African chimpanzees infanticide is almost unreported, and in most

other groups of East African chimpanzees occurs at much lower levels, so I would urge caution in the statement that rates in the wild are higher. It may also be worth highlighting that Lowe et al. suggest that one strategy for

countering infanticide in wild chimpanzees is to effectively absent themselves from the social group for weeks or months - a strategy that is not available for captive mothers (given as you note later the risks associated with

forced isolation).

  • I'm not sure whether or not there has been any consideration of increased risk of intrinsic disease when managing breeding of captive chimpanzees? Could this explain variation in rates with wild individuals.

  • I note the frequent use of the term 'AZA-accredited facilities' while I understand essentially what this means, and I realise this is a contribution to a special issue on welfare, I wonder if it might help a broader readership to have a brief sentence or two at the start of the methods outlining what AZA accreditation implies, and perhaps also what proportion of chimpanzees in the US are held outside of AZA accredited facilities.

  • Lines 356-360. I agree, however, it is unclear from your phrasing at the moment that conspecific aggression within communities in the wild is very much in the minority as compared to between communities. So I'm not sure that combining them and then presenting this in comparison to within group captive aggression is appropriate - perhaps just a little clarification needed (I think the Wilson et al. paper cited earlier provides a useful breakdown).

  • Line 372. Are there any data available about the number of individuals who sustained serious injuries in these types of events? As these may lead to death in the wild, but be treated in captive groups.

  • There is an extra 1. At the start of the references
